# ATP7B-Deficient Hepatocytes Reveal the Importance of Protein Misfolding Induced at Low Copper Concentration

**DOI:** 10.3390/cells11213400

**Published:** 2022-10-27

**Authors:** Peggy Charbonnier, Benoît Chovelon, Corinne Ravelet, Tuan Dung Ngo, Mireille Chevallet, Aurélien Deniaud

**Affiliations:** 1Université Grenoble Alpes, CNRS, CEA, IRIG Laboratoire de Chimie et Biologie des Métaux, F-38000 Grenoble, France; 2Service de Biochimie SB2TE, Institut de Biologie et Pathologie CHU Grenoble Alpes, F-38000 Grenoble, France; 3Département de Pharmacochimie Moléculaire, Université Grenoble Alpes, CNRS, UMR 5063, F-38041 Grenoble, France

**Keywords:** copper homeostasis, ATP7B, Wilson disease, hepatocytes, protein misfolding, CrispR/Cas9

## Abstract

Copper is a transition metal essential for human life. Its homeostasis is regulated in the liver, which delivers copper to the whole body and excretes its excess outside the organism in the feces through the bile. These functions are regulated within hepatocytes, and the ATP7B copper transporter is central to making the switch between copper use and excretion. In Wilson disease, the gene coding for ATP7B is mutated, leading to copper overload, firstly, in the liver and the brain. To better understand the role of ATP7B in hepatocytes and to provide a smart tool for the development of novel therapies against Wilson disease, we used the CrispR/Cas9 tool to generate hepatocyte cell lines with the abolished expression of ATP7B. These cell lines revealed that ATP7B plays a major role at low copper concentrations starting in the micromolar range. Moreover, metal stress markers are induced at lower copper concentrations compared to parental cells, while redox stress remains not activated. As shown recently, the main drawback induced by copper exposure is protein unfolding that is drastically exacerbated in ATP7B-deficient cells. Our data enabled us to propose that the zinc finger domain of DNAJ-A1 would serve as a sensor of Cu stress. Therefore, these Wilson-like hepatocytes are of high interest to explore in more detail the role of ATP7B.

## 1. Introduction

Copper (Cu) is a micronutrient essential for human life. At the organismal scale, its homeostasis is regulated in the liver, where hepatocytes absorb Cu arriving from the intestine. From the liver, Cu is then distributed via the blood, and in case of excess, Cu is excreted outside the body through the bile and, thereof, the feces. ATP7B, also named Wilson ATPase, is central in this regulation mechanism since the protein traffics between the Golgi apparatus to supply Cu for the maturation of proteins such as the ceruloplasmin and bile canaliculi to perform the excretion into the bile [1]. ATP7B is an integral membrane protein that transports Cu between both sides of a lipid bilayer in a directional way. This transport is energized by ATP hydrolysis. This part of Cu homeostasis is called the secretory pathway, and it also involves the Cu chaperone Atox1, a small cytosolic protein that complexes Cu in the Cu(I) state and delivers it to ATP7B. However, the mechanisms behind the efficiency of this Cu transfer process remain elusive. In the liver, only ATP7B is expressed, while in other tissues, ATP7A or both ATPases are expressed, such as in the brain, where a complex interplay between ATP7A and ATP7B enables the proper supply of Cu in the different brain areas [2].

The Wilson disease is an autosomal recessive rare genetic disorder in which the *atp7b* gene is mutated. Various mutations exist that trigger dysfunctions in ATP7B Cu transport, folding or intracellular trafficking [3]. Overall, Cu excretion to bile canaliculi is impaired leading to Cu accumulation in Wilson patients, firstly in the liver and in the brain (for a review on Wilson disease [4]). The evolution of the disease is long and depends on multiple and unclear factors [4,5]. Multiple functions are thus impacted, such as lipid metabolism [6,7,8], which explains the evolution toward steatosis [9]. Moreover, due to the central role of ATP7B in Cu homeostasis regulation in the brain, neurological disorders are associated with Wilson disease [10]. The current therapeutic options in Wilson disease have been unchanged since the sixties. The main treatments, D-Penicillamine (D-pen) and Trientine, are copper chelating drugs that promote copper excretion mainly into urine. However, the use of these chelators requires a life-long treatment leading to side effects that limit treatment compliance. More recently, bis-choline-tetrathiomolybdate (TTM), a copper chelator with high affinity for Cu(I), has been produced and successfully tested in phase II clinical trials, and is currently in phase III but its safety is not proven yet [11].

For decades, various animal models have been used, but they do not fully recapitulate the evolution of the disease and, in particular, the severity of the neurological disorders [12]. Furthermore, the use of animal models should be limited as much as possible in the next decades, and future achievements in the field of Cu homeostasis and Wilson disease must go toward integrative cell biology, ideally using organoid models. Indeed, they will enable us to perform analysis down to the subcellular level in a context mimicking an organ. Recently, we showed that the HepG2/C3a hepatocarcinoma cell line can be grown in 3D and spontaneously form bile canaliculi structures active for the excretion of organic molecules as well as Cu or silver [13,14]. This cellular system could thus be very interesting for further study related to Wilson disease. To reach this goal, we used the CrispR/Cas9 strategy to engineer the HepG2/C3a cell line by impairing the *atp7b* coding region. The response to Cu in Wilson hepatocytes was compared to the parental cell line. On the one hand, the sensitivity to a severe Cu stress is only decreased by about 100 µM, which corresponds to 20% of the lethal dose for 50% of the cells. On the other hand, the cellular responses occurring upon moderate Cu stress revealed the important role of ATP7B at a few micromolar Cu concentrations. Interestingly, metallothioneins (Met) expression and glutathione (GSH) production started at lower Cu concentration but reached levels in the same order of magnitude as in HepG2/C3a cells, while HSPA6 expression was drastically exacerbated in the ATP7B-deficient cells. These results highlight the importance of Cu-induced protein misfolding in the Wilson-like hepatocytes, while other stresses were not modified to that extent. Moreover, we propose that the DNAJ-A1 Zn finger serves as a sensor of Cu stress to initiate the HSPA6 response. Finally, we showed that the absence of ATP7B leads to the impairment of the farnesoid X-receptor (FXR) at micromolar Cu concentrations, which is consistent with previously published data on Wilson disease [6,8].

## 2. Materials and Methods

### 2.1. Generation of ATP7B Knock-Out HepG2/C3a Cell Lines

To obtain a cell line without expression of the ATP7B protein, we used the CrispR/Cas9 technology based on non-sense deletion or insertion within the open reading frame of the *atp7b* gene. The pLentiCrispRv2 developed by the group of Feng Zhang was used to do it. The general strategy was thus based on [15,16].

Different online tools were used to design sgRNA targeting the *atp7b* sequence. Four sequences were chosen: GCGGGGCGATATCGTCAAGG, GGCTGTGGTCAAGCTCCGGG, CTTAGATAAGATTTTCCGAC, and GATAAGTGATGACGGCCTCT named sg1, sg2, sg3, and sg4, respectively. The sgRNA sequences were cloned into the pLentiCrispRv2 by a restriction-ligation strategy. The pLentiCrispRv2 was cut using BsmB*I* and then dephosphorylated using Antarctic phosphatase. The primers containing the sgRNA sequences were phosphorylated using polynucleotide kinase and then annealed by ramping the temperature from 95 °C to 25 °C at 0.1 °C/s. The duplexes were then ligated into the digested pLentiCrispRv2 plasmid using the quick ligation kit from Roche. Ligation products were transformed into Stbl3-competent cells. Plasmids containing sgRNA were identified by restriction profiles and confirmed by sequencing.

HEK293T cells were used to obtain lentivirus able to express the CrispR/Cas9 machinery and the guide RNA of interest. HEK293T cells were transfected with pLentiCrispRv2-sgRNA*atp7b*, pMD2.G and psPAX2 using PEI [17]. The media was changed after 12 h for fresh DMEM media supplemented with 10% fetal bovine serum (FBS). After 48 h, the media containing the viruses was recovered, centrifuged 10 min at 3000 rpm to pellet cell debris. The supernatant that contains the viruses was stored protected from light at 4 °C until use.

To perform HepG2/C3a cells engineering by the different viruses, cells were plated in 24-well plate at a density of 2.5 × 10^4^ cells per well. For each virus 10, 100 or 1000 µL of virus was added in different wells. After 24 h of incubation with the viruses, media was changed for DMEM supplemented with 10% FBS and 1 µg/mL of puromycin. Forty-eight hours later, cells were recovered diluted at 5 cells per mL. This diluted cell suspension is then used to seed 96-well plates with 100 µL per well. The growth of isolated clones was then scrutinized in the following days. Isolated clones were expanded for further genotypic analyses.

The characterization of the *atp7b* coding sequence in the different clones was performed using a 2-step strategy. In a first step, the region surrounding the location of the sgRNA sequence was amplified by PCR using two sets of primers in order to be sure to amplify this region whatever the size of the possible insertion or deletion generated following Cas9 cleavage. The sequences of the different pairs of primers can be found in Appendix A. Moreover, the amplicons were separated on acrylamide gels to obtain a resolution enabling to discriminate PCR fragments with few nucleotide differences. For the clones having promising migration profiles (Appendix A as an example), the PCR products were ligated into Topo plasmid (ThermoFisher Scientific, Courtaboeuf, France) and transformed into Top10 competent cells. For each HepG2/C3a-derived clones, plasmid DNA of about 10 bacterial clones were purified and sequenced using M13 primers. Based on sequencing data, HepG2/C3a-derived clones with non-sense insertion or deletion in the *atp7b* region in both alleles were selected.

### 2.2. Cell Culture

HepG2/C3a were provided by the ATCC. HepG2/C3a cells were grown in MEM media supplemented with 10% FBS and 1% antibiotics-antimycotic at 37 °C in a humidified incubator and with an atmosphere of 5% CO_2_. Cells were exposed to Cu at the indicated concentrations and for the indicated times. When required, chenodeoxycholic acid (CDCA, Sigma-Aldrich, Saint Quentin Fallavier, France) at 80 µM or the equivalent volume of DMSO was added for the last 17 h of exposure.

### 2.3. Immunofluorescence

The different cell lines were grown on glass coverslips until approximately 60% confluence. The cells were then incubated overnight with 1, 10, or 100 µM copper chloride solution (CuCl_2_) or with 200 µM bathocuproine sulfonate (BCS) in growth medium. After the incubation, cells were washed in PBS and fixed for three minutes with cold methanol on ice. After fixation, cells were rinsed three times with PBS. Cells were then permeabilized with 0.5% Triton X-100 in Tris-Buffered Saline with 0.05% Tween-20 (TBST) for around 15 min at room temperature and then blocked with 0.1% Triton X-100, 3% BSA in TBS for 1 h at room temperature. For immunostaining, the cells were incubated with antibodies against ATP7B (Abcam ab124973, 1:50), Golgin-97 (A21270, 1:100, ThermoFisher Scientific, Courtaboeuf, France), MRP2 (ab3373, 1:50, abcam, Paris, France), or LAMP1 (sc20011, 1:50, Santa Cruz Biotechnology, Heidelberg, Germany) in antibody buffer (0.1% triton X-100/1% BSA in TBST) for 1 h at room temperature. After three washes in PBS, cells were then incubated with secondary antibodies diluted 1:1000 (Alexa Fluor 488 goat anti-rabbit antibody and Alexa Fluor 568 goat anti-mouse antibody, ThermoFisher Scientific, Courtaboeuf, France) in antibody buffer for 1 h at room temperature. The cells were then rinsed in PBS and incubated for 5 min in the last wash with Hoechst diluted 1:1000 prior to being mounted with Mounting medium (Sigma-Aldrich, Saint Quentin Fallavier, France) onto a glass microscope slide. Cells were finally imaged using Zeiss LSM880 inverted laser scanning confocal microscope with a 63× oil immersion objective.

### 2.4. Western Blot

HepG2/C3a cell line and its derived clones without ATP7B expression were seeded at 1.2 × 10^6^ cells in 6-well plates (CellClear Greiner) in MEM media supplemented with 10% FBS. After overnight culture, cells were washed one time with PBS to remove cellular debris before incubation with complete medium containing Cu or BCS at the indicated concentrations for 24 h. Harvested cells were centrifuged at 5000× *g* for three minutes, and the pellets were washed two times with PBS before resuspension in cold RIPA cell lysis buffer (100 mM HEPES pH 7.4, 1% Triton X-100, 0.5% deoxycholate) supplemented with protease inhibitors for 20 min on ice. The cellular debris was then eliminated by centrifugation at 15,000× *g* for 30 min, and the supernatant fractions were analyzed by Western blot with antibodies against Atox1 (sc-100557, Santa Cruz Biotechnology, Heidelberg, Germany) and ATP7B (ab124973, abcam, Paris, France). Equal amounts of loaded proteins were previously quantified using a Micro BCA assay kit (ThermoFisher Scientific, Courtaboeuf, France).

### 2.5. RNA Extraction and Quantification

At the indicated time points, cells were harvested, and mRNA were isolated using the Nucleospin RNA kit (Macherey-Nagel). The RNA concentration was determined using a NanoDrop spectrophotometer (ND-1000). RNA were reverse transcribed with the Maxima First Strand cDNA synthesis kit (ThermoFisher Scientific, Courtaboeuf, France) according to the manufacturer’s instructions using random primers. Quantitative PCR was performed as described in [18] using the PowerUp^TM^ SYBR^TM^ Master Mix (ThermoFisher Scientific, Courtaboeuf, France). qPCR reactions were run in triplicate, and quantification was performed using comparative regression (Cq determination mode) using Cfx (Bio-Rad Cfx manager version 3.0, Hercules, CA, USA) with GAPDH and HPRT amplification signals as housekeeping genes to correct for total RNA content, while the untreated sample was used as the “calibrator”. At least three biological replicates were performed for the presented data. The primers used are described in Appendix A.

### 2.6. FXR Activity

The activity of the FXR nuclear receptor was determined by measuring the amount of one of its target transcripts (BSEP) upon the addition of a specific ligand of the receptor. This activity was compared in hepatocytes exposed or not to Cu in order to measure the impact of Cu on FXR activity. *CDCA* was used to induce the expression of *bsep* (coding for Bile Salt Export Pump, BSEP) [8]. The control (*ctl*) sample (unexposed to Cu) was set to 100% activity of FXR, and it corresponds to the *ratio* (1):(1)$$Ratio\left(ctl\right)=\frac{mRNA\;level\;with\;ligand\;\left(CDCA-HC\right)}{mRNA\;level\;without\;ligand}$$

For all conditions of Cu exposure, we similarly calculated the *ratio* (2):(2)$$ratio\left(condition\;Z\right)=\frac{mRNA\;level\;with\;ligand\;\left(CDCA-HC\right)}{mRNA\;level\;without\;ligand}$$

In order to determine the percentage of activity in *condition Z*, we applied the following Equation (3):(3)$$\%\;activity\;Z=\left[\frac{\left(Ratio\left(conditionZ\right)-1\right)}{\left(Ratio\left(ctl\right)-1\right)}\right]\times 100$$

Subtracting 1 takes into account that the value of a control in mRNA normalized relative expression is 1.

### 2.7. ICP-MS

Cu was quantified using a quadrupole ICP-MS (Perkin Elmer NexION 2000, Waltham, MA, USA). The collision cell technology (CCT) was used with only Helium.

Cells exposed to different concentrations of Cu were mineralized at atmospheric pressure in 67% HNO3, for 24 h at room temperature then 24 h in the oven at 60 °C. The mineralization was diluted 1:100 before analysis. Standard solutions were prepared in nitric acid 1% (*v*/*v*). ^65^Cu was measured and ^89^Y at 472 nmol/L was used as internal standard.

### 2.8. Statistical Analysis

All the quantitative data presented with a standard deviation are based on at least three biological replicates. Statistical comparisons are based on a one-way ANOVA test.

## 3. Results

### 3.1. Generation of ATP7B-Deficient Cell Lines Deriving from the HepG2/C3a Cell Line

The HepG2/C3a cell line was derived from HepG2 [19] but possessed properties closer to primary hepatocytes in terms of metabolism. For instance, we recently showed that the HepG2/C3a cell line, grown as spheroids, forms active bile canaliculi with a high density of microvilli [13], while the HepG2 cell line does not. Biliary excretion properties are central to work on ATP7B function since this protein traffics between the Golgi apparatus and bile canaliculi depending on the Cu level in hepatocytes. We also showed that this mechanism is fully active in HepG2/C3a cell line, in particular grown as a spheroid [13]. This explains our choice for the HepG2/C3a cell line to generate cell lines deficient in ATP7B using the CrispR/Cas9 methodology.

Different tools have been used to design sgRNA sequences in order to generate random deletions or insertions in the gene coding for ATP7B. To maximize the chances of success four different sgRNA (see Materials and Methods section) have been tested but only two led to the production of cell lines with modifications in the targeted gene. The isolated clones were screened via a two-step strategy (see details in the Materials and Methods section). Firstly, the region targeted by the sgRNA was amplified by PCR and the products were separated on an acrylamide gel to analyze amplicon size at high resolution. This strategy enabled to identify small insertions or deletions by comparison with the PCR products generated using the parental cell line (an example is presented in Appendix A). The strength of this methodology resides in the capacity to detect even very small deletions of few nucleotides and to analyse both alleles at once, which is crucial to be sure to avoid keeping one allele in the wild-type form. In a second step, for selected clones based on step one, PCR products of the targeted region were subcloned into Topo cloning plasmid, and transformed in *E. coli*. Plasmids contained in different *E. coli* clones were amplified and sequenced. This approach enables to obtain the exact sequences of both alleles of the targeted gene for each cell line tested, which is required to know whether both alleles are mutated with non-sense deletion or insertion.

Several clones were thus selected for further analysis in terms of ATP7B protein expression and sensitivity to Cu. The clones named sg1-1, sg2-1, and sg2-2 contain non-sense deletions for both alleles and come from two different sgRNA (sg1 and sg2). The clone sg1-2 contains one non-sense deletion and a second allele with a deletion of 111 nucleotides leading to the in-phase deletion of 37 amino acids in ATP7B. The clone sg1-3 is not mutated in the *atp7b* gene and is therefore used as a control of possible off-target effects with sg1. Western blot analysis confirmed the absence of ATP7B expression in the clones sg1-1, sg2-1, and sg2-2 and the weak expression for sg1-2, while sg1-3 properly expresses ATP7B (Figure 1A).

Expression analysis was further confirmed by immunofluorescence analysis of ATP7B. In the initial HepG2/C3a cell line, ATP7B is localized in the Golgi apparatus upon Cu-deprived conditions (Figure 1B, incubation with the Cu chelator BCS) and at bile canaliculi upon exposure to 100 µM Cu (Figure 1C), while basal conditions led to intermediate location (Figure 1D). Similar immunofluorescence analyses performed in BCS and Cu conditions on the cell lines sg1-1 (Figure 1E) and sg1-3 (Figure 1F) confirmed the absence of ATP7B in the former, while the latter behaves as the parental cell line. For further functional analysis, sg1-1 and sg2-1 cell lines were chosen as models for ATP7B-deficient cell lines, but similar data have been obtained with sg2-1 and sg2-2 cell lines.

### 3.2. Atox1 Expression

Cu trafficking to the secretory pathway involves the Atox1 protein upstream of ATP7B. Its expression was thus compared in the different cell lines isolated and in different conditions, i.e., in presence of Cu supply or deprivation (Figure 2). In the HepG2/C3a cell line, the expression of Atox1 did not change upon Cu exposure but it is slightly increased upon deprivation using the BCS Cu chelator. In the two cell lines without ATP7B expression we selected, sg1-1 and sg2-1, the expression of Atox1 is slightly increased compared to HepG2/C3a cells in basal conditions and there is no variation induced neither by Cu exposure nor by Cu depletion. Overall, these data showed that the absence of ATP7B did not have a significant impact on Atox1 expression. On the contrary, it was previously shown that Atox1 would play an important and complex role for ATP7B function enabling Cu exchange in both directions [20]. Moreover, it was previously published that Atox1 knock-out impact the Cu-dependent localization of ATP7A [21]. Therefore, both Cu and Atox1 would act on ATP7A and ATP7B trafficking, while a reciprocal effect from ATP7B to Atox1 must not be the case.

### 3.3. Sensitivity to Cu and Effect of Cu Chelator

The sensitivity to Cu stress was tested for these different clones by the determination of the Cu dose lethal for 50% of the cells (Cu LD50) (Figure 3A). For the initial HepG2/C3a cells, the Cu LD50 is roughly 470 µM, which is similar to the non-mutated sg1–3 cell line meaning that the CrispR process did not induce an impact on Cu sensitivity for these cells, precluding off-target effects. However, the four other cell lines have Cu LD50 between 340 and 400 µM, which is significantly more sensitive to Cu compared to the cell lines with WT ATP7B. Therefore, these experiments confirmed that the clones mutated in the *atp7b* gene possess an increased sensitivity to Cu but remains able to deal with relatively high amount of extracellular Cu concentration up to 300–350 µM. The protection against Cu stress by standard Wilson disease therapeutics, i.e., D-Pen, trientine, and TTM, has been assessed on the parental HepG2/C3a cell line and two complete *atp7b* KO cell lines, sg1-1 and sg2-1. All cell lines tested were significantly protected by the chelators (Figure 3B). The three molecules led to an increase of ~200 µM in Cu LD50 for the HepG2/C3a cell line, while the protection with the three chelators was different for the sg1-1 and sg2-1 *atp7b* KO cell lines. The three chelators ranked in the order TTM, D-Pen, and then trientine, providing an increase in Cu LD50 of ~100, 150–200, and 250–300 µM, respectively (Figure 3B). TTM is, therefore, less active in ATP7B-deficient cells, maybe because its action is favored by the presence of this Cu transporter.

Based on these results, one can make the hypothesis that several mechanisms act to protect the cells against Cu stress, and the range of Cu concentration cells are exposed to trigger different cellular response mechanisms. To better understand them in cell lines with or without ATP7B, intracellular Cu was measured upon exposure to increasing concentrations of Cu (Figure 4A). In HepG2/C3a cells, intracellular Cu reached the first plateau around 0.4 nmol per million cells when they were exposed to 2 to 10 µM Cu. At 100 µM Cu, the intracellular concentration is only twice. The analysis of ATP7B trafficking in HepG2/C3a cells upon exposure to Cu concentration between 1 and 100 µM showed a significant localization at bile canaliculi (Figure 4B and Figure 1C) starting at 1 µM, and a full localization was observed at 100 µM. In ATP7B-deficient cells, intracellular Cu reached a plateau at slightly lower values than HepG2/C3a cells, but interestingly, exposure to only 2 µM Cu is sufficient to reach this level of intracellular Cu. Overall, these results showed that, upon exposure to moderate concentration of Cu (below 100 µM), hepatocytes initiate two mechanisms. First, Cu excretion thanks to ATP7B trafficking to bile canaliculi at micromolar concentration. Second, Cu entry reduction thanks to Ctr1 endocytosis [22,23], which should play a major role at concentrations higher than 10 µM. In addition to these classical Cu homeostasis mechanisms, hepatocytes can trigger different stress pathways upon Cu exposure. These have been analyzed in detail in the different cell lines of interest by mRNA quantification.

### 3.4. Metal Stress Response Started at Low Cu Concentration in Cell Line with ATP7B

Met are small cysteine-rich proteins involved in the chelation of excess metal ions such as Zn(II), Cu(I), Cd(II), or Ag(I) (for review [24]) and the glutamate cysteine ligase 1 (GCLM) is the rate-limiting enzyme involved in GSH synthesis (for review [25]). GSH is crucial to maintain the intracellular environment in a reduced state and could also play a role in Cu(I) chelation. Therefore, monitoring the expression of Met and GCLM is often used as sensors of metal overload in the cell. Herein, their mRNA level was determined after exposure for 6 h to Cu concentrations between 2 and 200 µM (Figure 5A,B). These concentrations have been chosen based on the effect of Cu on ATP7B migration that is initiated at 1 µM Cu and maximal at concentrations equal to or higher than 100 µM (Figure 1B–D and Figure 4B). For the WT HepG2/C3a cell line, Met expression drastically increases at Cu concentration higher than 10 µM reaching between 20 and almost 100-fold increase for 200 µM Cu exposure, while GCLM is slightly increased upon Cu exposure up to 4-fold at 200 µM Cu (Figure 5A,B). For the two complete *atp7b* KO cell lines, sg1-1 and sg2-1, Met expression increases more than 20-fold at only 2 µM Cu, showing that these cells had to cope with Cu overload for very limited Cu concentration exposure. However, at concentrations equal to and above 50 µM Cu, similar Met responses were observed for the HepG2/C3a and sg2-1 cell lines, while Met is slightly more expressed for the sg1-1 cell line (Figure 5A). In addition, GCLM induction is higher, with the ATP7B-deficient cell lines reaching a 12- to 19-fold increase with the sg1-1 and sg2-1 cell lines, respectively (Figure 5B). Altogether, these results are consistent with Cu accumulation data (Figure 4A) and confirm the crucial role of ATP7B for the regulation of low Cu concentrations exposure up to 10–50 µM that could be compensated by Met overexpression and GSH production in the *atp7b* KO context. Moreover, GSH production seems more important for the ATP7B-deficient cell lines at Cu concentrations higher than 10 µM, which is probably critical to maintaining a properly reduced potential in these cells. To assess the evolution of the Met response over time, its expression has been determined after 24 h exposure for Cu concentrations between 2 and 40 µM (Figure 5C). In all cases, Met level remained similar to those observed after 6 h exposure showing that Met response is fast and sustained.

### 3.5. Redox Stress Is not Central upon Cu Exposure

Excess of intracellular Cu is often considered a potential source of oxidative stress in the cell. The expression of genes associated with enzymes involved in the conversion of reactive oxygen species (ROS) has been followed in the different cell lines exposed to increasing concentrations of Cu. The two main enzymes involved are superoxide dismutase (SOD) (Figure 5D) and catalase (Figure 5E). Their expression did not vary a lot depending on Cu exposure. However, there was a slight increase in the expression of SOD at 200 µM Cu for the sg1-1 cell line. Catalase was downregulated to a greater extent, down to about 1.7-fold for the HepG2/C3a and the sg2-1 cell lines and more than two-fold for the sg1-1 cell line. These downregulations started at the lowest concentration of 2 µM Cu for the KO cell lines, while they occurred at higher Cu concentrations for the WT cell line. These data confirmed the sensitivity of the ATP7B-deficient cell lines to low Cu concentrations, but the downregulation of catalase did not fit with a model involving ROS production upon Cu overload. In parallel, the production of intracellular ROS was determined using a cell-permeable and trapped version of H_2_DCFDA (Figure 5F). HepG2/C3a, sg1-1, and sg2-1 cell lines followed the same trend with a slight but significant increase in ROS production upon exposure to concentrations of Cu higher than 100–150 µM reaching about a 2-fold increase. Overall, the data showed that Cu excess did not lead to important ROS production even in the context where the major Cu exporter in hepatocytes (ATP7B) is absent, while heavy metal stress is clearly observed in the cells, which is more pronounced in ATP7B-deficient cell lines.

### 3.6. Protein Misfolding upon Cu Exposure Is Exacerbated in ATP7B-Deficient Cell Lines

Heme oxygenase 1 (HMOX) is frequently upregulated upon heavy metal exposure [18,26,27]. It is a mediator of the Nrf2 stress pathway and would be involved in response to various stresses, including inflammation [28,29]. In the different cell lines, HMOX expression gradually increased with Cu concentration (Figure 6A). While the effect is similar up to 10 µM Cu, the overexpression is more than 3-fold higher at 50 µM Cu for the two KO cell lines, which is similar to the HMOX level at 200 µM Cu for the WT cell line. These data confirmed the activation of HMOX1 upon Cu exposure. However, in the ATP7B-deficient cell lines, a delay was observed in the activation of this pathway compared to Met and GCLM, starting at 10–50 µM Cu and 2–10 µM Cu, respectively, while the upregulation of these three genes started at 10 µM Cu for the HepG2/C3a cell line. Since HMOX is involved in inflammatory response, IL8 was tested as well (Figure 6B). A moderate increase was observed for the HepG2/C3a cell line starting at 10 µM Cu and reaching 10-fold increase at 200 µM Cu. The sg1-1 cell line presented the highest increase up to more than 50-fold, while sg2-1 only reached about 16-fold increase. The response to Cu was stronger for the ATP7B-deficient cell lines that expressed 3 times more IL8 mRNA than the WT cell line at only 10 µM Cu.

Protein unfolding was recently proposed as an important effect of Cu overload [30]. The importance of this pathway in the different cell lines was tested by analyzing the expression of HSPA6 (Figure 6C), which is an inducible chaperone belonging to the HSP70 family and possessing the capability to refold proteins thanks to its ATPase domain. As expected, HSPA6 is strongly overexpressed in the HepG2/C3a cell line upon Cu exposure, starting from an about 2-fold increase at 2 µM Cu and reaching almost 150-fold at 200 µM Cu. However, in ATP7B-deficient cell lines, the impact is an order of magnitude higher or more depending on the conditions used. The sg1-1 and sg2-1 cell lines started with a ~10- and ~90-fold increase for 2 µM Cu and ended up at almost 1500- and 2500-fold for 200 µM Cu, respectively. These data indicate the dramatic role of Cu overload on protein unfolding, even at low Cu concentration, and clearly highlight the impact of the absence of ATP7B on this process, which is, therefore, the main problem in the context of Wilson disease. To identify a specific pathway that could be involved in sensing Cu-induced protein unfolding, the expression level of chaperones belonging to the HSP40 family, such as DNAJs, in humans was determined. DNAJ-A1, DNAJ-A2, and DNAJ-A3 were assessed to try to identify a specific pathway for Cu sensing. These three proteins were chosen since they contain zinc finger domains that are known to be used as sensors of the redox potential, for instance. It is also known that Cu(I) can displace Zn(II) from the Zn finger [8,31], and we made the hypothesis that it could play a role as Cu(I) sensor since the affinity of Cu(I) for these motifs is in the order of Cu-chaperones binding site. Moreover, DNAJ-A1 and DNAJ-A2 are cytosolic, while DNAJ-A3 is mitochondrial, giving the possibility to assess this putative role in different subcellular compartments. For DNAJ-A2, the tendency is the same for the three cell lines, with a very slight increase at Cu concentrations higher than 50 µM and reaching a maximum of 1.7-fold (Figure 6E). For DNAJ-A3, a similar slight increase was observed for the WT cell line, while the KO cell lines inversely showed a low decrease down to 1.7-fold (Figure 6F). The most significant effect was revealed with DNAJ-A1, which expression was significantly increased at 10 µM Cu and higher concentrations or even at 2 µM for the sg2-1 cell line (Figure 6D). In this case, the expression reached a higher level for the KO cell lines compared to HepG2/C3a cells up to 3-fold and 2-fold, respectively. Interestingly, the thresholds for HSPA6 and DNAJ-A1 upregulation are correlated since a 2-fold upregulation of DNAJ-A1 corresponds to a steep increase in HSPA6 expression for all the cell lines. Therefore, our data revealed for the first time that DNAJ-A1 could play a role in sensing Cu(I) overload in human cells since it is overexpressed upon Cu exposure, and this effect is higher in a context where ATP7B is not expressed. A hypothesis that needs to be further investigated in the future since it could play a central role in Cu signaling and homeostasis regulation.

### 3.7. Autophagy Is not Activated upon Cu Exposure in the Different Hepatocyte Cell Lines

Finally, the expression of genes involved in autophagy has been assessed since it has been described that autophagy mechanisms would be activated in Wilson disease to protect hepatocytes from Cu-induced apoptosis [32]. Although many genes (ATG7, ATG13, BNIP3L, FUNDC1, Get3, PINK1, and Drp1) involved in autophagy and mitochondrial autophagy, also named mitophagy, were assessed in HepG2/C3a, and the two ATP7B-deficient cell lines (Figure 7), only minor, non-concentration-dependent, and mainly non-significant variations were observed at Cu concentrations up to 200 µM. The only exception is Get3 which is significantly downregulated in the sg1-1 cell line upon exposure to increasing concentrations of Cu between 10 and 200 µM. Altogether, one can imagine that the activation of autophagy mechanisms is at play upon long-term overload to Cu, where various disruptions in hepatocytes also occur.

### 3.8. Nuclear Receptor Activity Is Dramatically Impaired in the Absence of ATP7B upon Cu Exposure

Among the various effects observed in Wilson disease, it has been established in the last years that Cu can displace Zn from Zn finger DNA binding domains of specific nuclear receptors and, in particular, the farnesoid X-receptor (FXR) and the liver X-receptor. This metal exchange leads to the inhibition of the nuclear receptor and would be the cause of lipid homeostasis disruptions and, thereof, steatosis, a general process occurring in Wilson disease pathophysiology over the years [4,8,9]. A strategy based on the quantification of the mRNA of target genes of these nuclear receptors has been developed to assess the inhibition of specific nuclear receptors [8,33]. For FXR, it is thus possible to quantify the production of mRNA coding for BSEP upon exposure to CDCA, which is known to activate the transcription activity of FXR. Using this strategy, we observed ~20%, ~30%, and ~75% inhibition of FXR activity in the HepG2/C3a cell line exposed to 2, 10, and 40 µM Cu, respectively (Figure 8). In the ATP7B-deficient cell lines, a drastic inhibition of FXR was revealed at only 2 µM Cu. These data are very interesting since it shows that the impact of Cu on nuclear receptors is stronger in the absence of ATP7B, which contributes to favor the development of this steatosis process in Wilson disease patients. Moreover, our findings completed an overview of the Cu(I) binding to Zn finger motifs involved in both the disruption of FXR and the activation of DNAJ-A1. These two processes are initiated at the lowest Cu concentration between 2 and 10 µM, depending on the cell line. Therefore, Zn fingers are the first chelators of Cu(I) in the cell, together with specific Cu chaperones. Moreover, Zn fingers can either serve as sensors to activate the stress response pathways as with DNAJ-A1 and HSPA6, or they can be functionally altered by Cu(I) binding as for FXR.

## 4. Discussion

In the current study, we have been able to develop novel HepG2/C3a-derived cell lines without ATP7B expression. Two different sgRNA led to ATP7B-deficient cell lines. Their characterization showed similar behavior upon moderate to acute Cu stress, which strengthened the conclusions that can be drawn with these novel Wilson-like cell lines. Moreover, their effects on the FXR pathway confirmed that these cell lines could reproduce disorders observed in Wilson patients, i.e., steatosis. Therefore, they are an interesting model for the study of Wilson pathophysiology or drug screening. Moreover, they can be grown in 3D culture to be closer to the liver architecture and physiology, which will be of paramount importance in the future to obtain reliable data without the use of animals.

The comparative analysis of the cellular responses to moderate Cu concentrations between 1 and 100 µM of the different cell lines also provided clues on Cu homeostasis mechanisms implementation in hepatocytes. Based on ATP7B trafficking experiments and Cu uptake quantification, it appears that ATP7B is the first player involved at low micromolar Cu concentrations to excrete this ion into bile canaliculi. Since Cu uptake reaches a similar level at a high Cu concentration, the regulation of Cu entry by Ctr1, together with the expression of Met to chelate Cu excess, is involved in a second step in the range of 10 to 100 µM Cu. All these processes are initiated at lower Cu concentrations in the absence of ATP7B. In terms of the stress response, our data clearly showed that protein misfolding is the main stress induced by Cu, which is drastically exacerbated in ATP7B-deficient cells. Indeed, our data showed a huge overexpression of HSPA6 that is known to be expressed under severe stress conditions [34,35,36]. Therefore, acting on protein unfolding represents a promising alternative to developing new therapeutic approaches for Wilson patients. Furthermore, we made the hypothesis that the Zn finger domain of DNAJ-A1 is a sensor of Cu(I) excess that would trigger the HSPA6 response. Actually, DNAJ proteins, also known as HSP40, are partner proteins of the HSP70 protein, to which HSPA6 belongs. HSPA6 is barely expressed in physiologic conditions, while HSP40 proteins are constitutively expressed and only slightly overexpressed in stress conditions [34]. Therefore, DNAJ-A1 could detect Cu(I) stress and triggers HSPA6 overexpression. However, this hypothesis remains to be confirmed, and it could also be interesting to assess the possible involvement of this Zn finger sensor for other Cu homeostasis regulation mechanisms.

In conclusion, we developed relevant cell lines for Wilson disease studies. Their comparison with the parental cell lines possessing ATP7B enables us to pinpoint the specific role of this protein upon Cu supply or deprivation. Therefore, these ATP7B-deficient cell lines are also interesting tools to better understand Cu homeostasis in hepatocytes.

## Figures and Tables

**Figure 1 cells-11-03400-f001:**
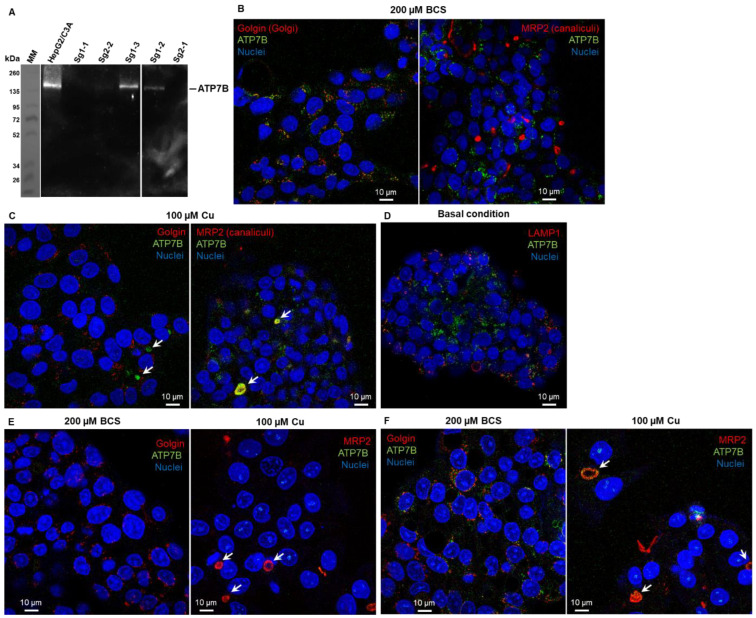
Characterization of CrispR/Cas9-modified HepG2/C3a cell lines. (**A**) Western blot analysis of the expression of ATP7B in the HepG2/C3a cell line and in different clones isolated after the CrispR/Cas9 process. Confocal fluorescence microscopy revealed that ATP7B (green) is localized in the Golgi apparatus (red) in HepG2/C3a cells in Cu-deprived conditions (exposure to the Cu chelator BCS) (**B**) and localized at bile canaliculi (white arrows) upon Cu exposure (100 µM) (**C**). In basal Cu conditions, ATP7B is in an intermediate location (**D**). Immunofluorescence analysis confirmed the absence of ATP7B in the sg1-1 cell line (**E**) and its presence in the sg1-3 cell line (**F**).

**Figure 2 cells-11-03400-f002:**
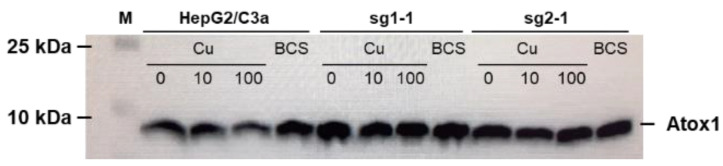
Expression of Atox1 in the different cell lines and depending on Cu conditions. Western blot analysis of the expression of Atox1 in the HepG2/C3a cell line and in different clones isolated after the CrispR/Cas9 process. Cells have been exposed to nothing, 10 µM Cu, 100 µM Cu, or 200 µM BCS for 24 h.

**Figure 3 cells-11-03400-f003:**
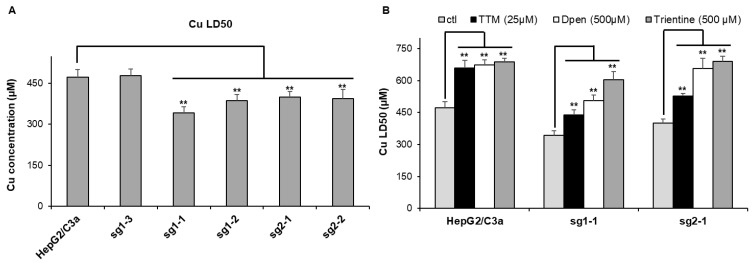
Sensitivity to Cu of the different cell lines. (**A**) Determination of the Cu LD50 for the HepG2/C3a cell line and for different clones isolated after the CrispR/Cas9 process. (**B**) Impact of various chelators on the lethality induced by Cu in the HepG2/C3a, sg1-1, and sg2-1 cell lines. The Cu LD50 was determined by MTT assay in the presence of the different compounds at the indicated concentrations. Data presented correspond to the average of at least three independent experiments +/− standard deviation. ** stands for data statistically different from the control (ctl) or from the HepG2/C3a cell line with *p* < 0.01.

**Figure 4 cells-11-03400-f004:**
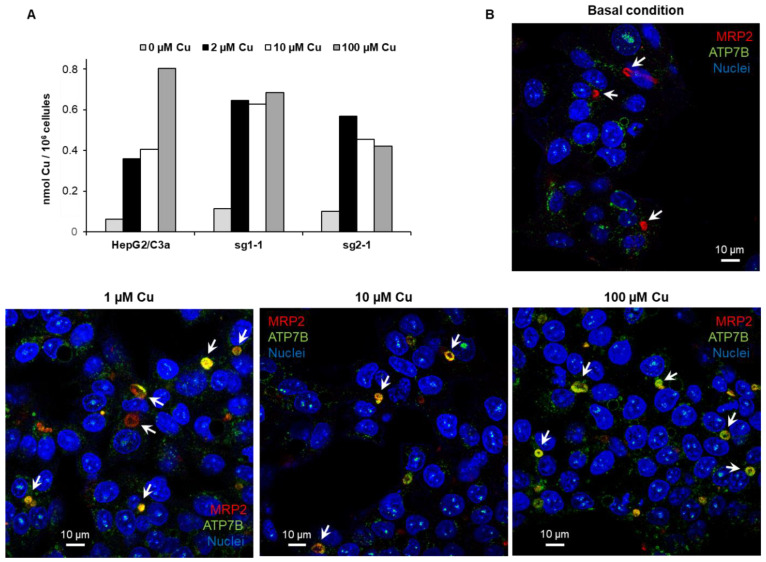
Cu accumulation in the different cell lines and ATP7B trafficking. (**A**) Amount of intracellular Cu for the HepG2/C3a, sg1-1, and sg2-1 cell lines upon exposure to increasing concentrations of Cu. (**B**) Analysis of ATP7B (green) localization by immunofluorescence in HepG2/C3a cells exposed to different concentrations of Cu. ATP7B is green, MRP2, a marker of bile canaliculi, is red, and nuclei are blue. White arrows pinpoint bile canaliculi.

**Figure 5 cells-11-03400-f005:**
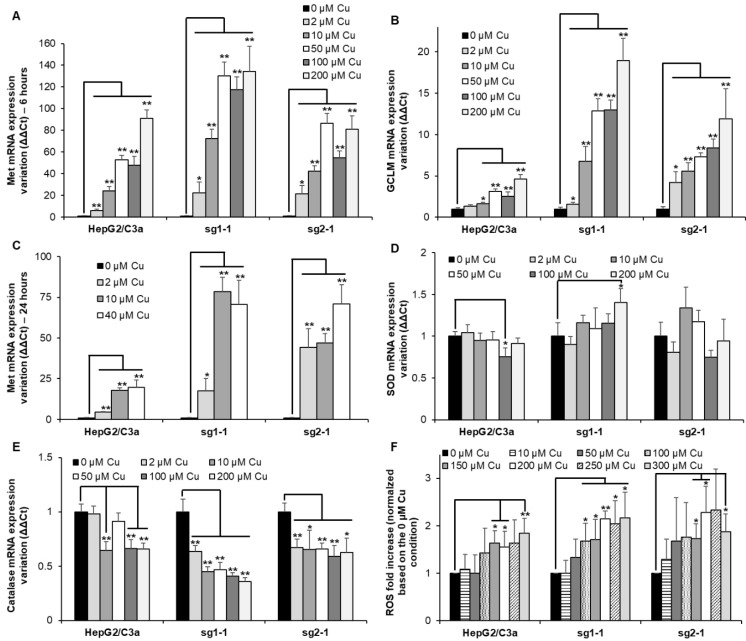
Metal and redox stress responses in cell lines with or without ATP7B. HepG2/C3a, sg1-1, and sg2-1 cells were exposed to increasing concentrations of Cu. Met (Met1X) (**A**), GCLM (**B**), SOD1 (**D**), and catalase (**E**) mRNA expression variations were determined following 6 h of exposure to Cu. Met1X (**C**) mRNA expression variation was also determined following 24 h of exposure to Cu. Results are expressed as the relative change in expression compared to the control. (**F**) ROS production determined by flow cytometry after exposure to Cu for 24 h and using the H_2_DCFDA probe following manufacturer instructions (ThermoFisher Scientific, Courtaboeuf, France). All the data are expressed as means +/− standard error of the mean of at least three independent experiments. * stands for data statistically different from the control unexposed to Cu with *p* < 0.05, and ** stands for data statistically different from the control unexposed to Cu with *p* < 0.01.

**Figure 6 cells-11-03400-f006:**
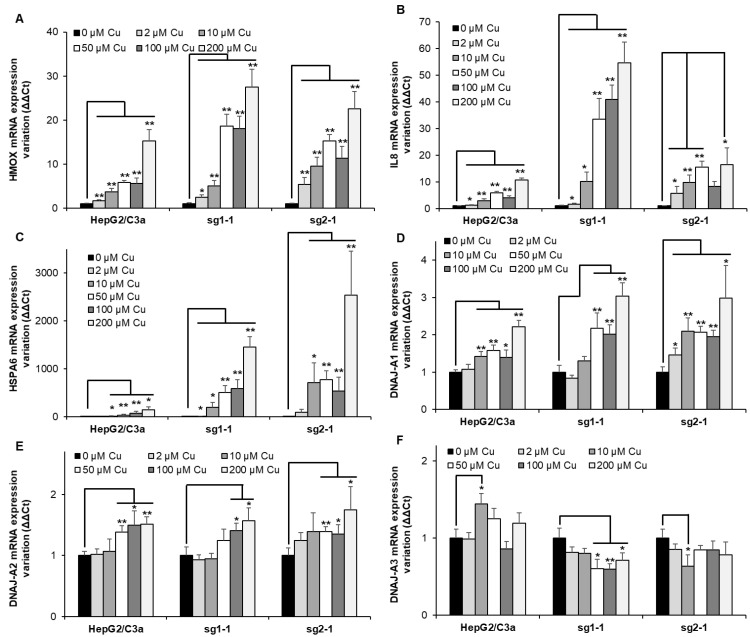
Inflammatory responses and protein misfolding in cell lines with or without ATP7B. HepG2/C3a, sg1-1, and sg2-1 cells were exposed to increasing concentrations of Cu. HMOX (**A**), IL8 (**B**), HSPA6 (**C**), DNAJ-A1 (**D**), DNAJ-A2 (**E**), and DNAJ-A3 (**F**) mRNA variation expression were determined following 6 h of exposure to Cu. Results are expressed as the relative change in expression compared to the control. All the data are expressed as means +/− standard error of the mean of at least three independent experiments. * stands for data statistically different from the control unexposed to Cu with *p* < 0.05, and ** stands for data statistically different from the control unexposed to Cu with *p* < 0.01.

**Figure 7 cells-11-03400-f007:**
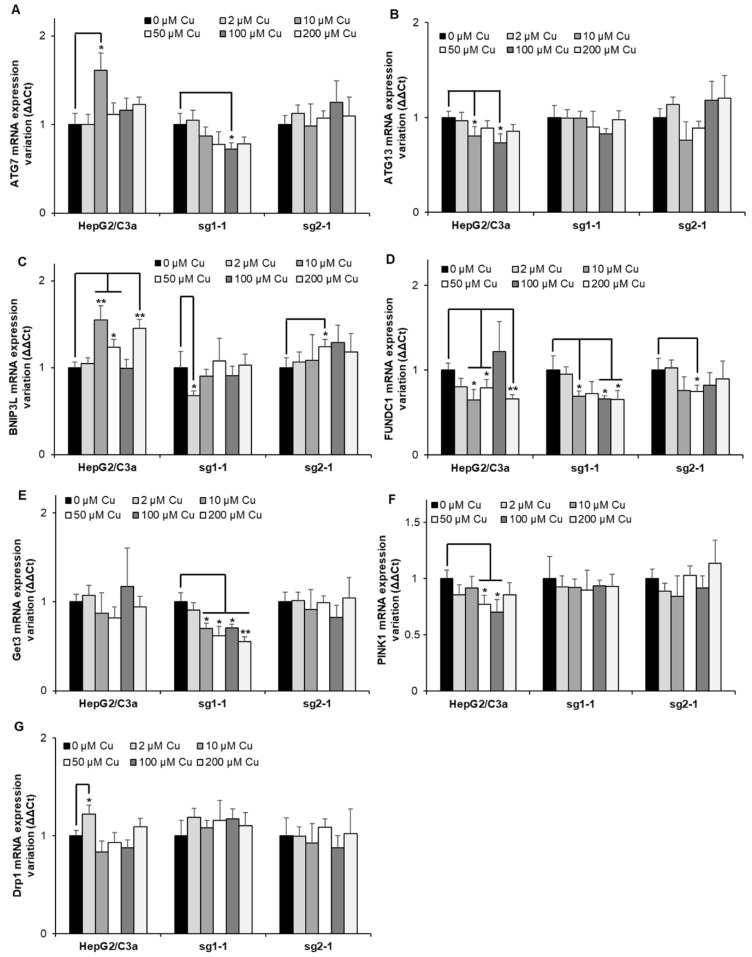
Autophagy response in cell lines with or without ATP7B. HepG2/C3a, sg1-1, and sg2-1 cells were exposed to increasing concentrations of Cu. ATG7 (**A**), ATG13 (**B**), BNIP3L (**C**), FUNDC1 (**D**), Get3 (**E**), PINK1 (**F**), and Drp1 (**G**) mRNA variation expression were determined following 6 h of exposure to Cu. Results are expressed as the relative change in expression compared to the control. All the data are expressed as means +/− standard error of the mean of at least three independent experiments. * stands for data statistically different from the control unexposed to Cu with *p* < 0.05, and ** stands for data statistically different from the control unexposed to Cu with *p* < 0.01.

**Figure 8 cells-11-03400-f008:**
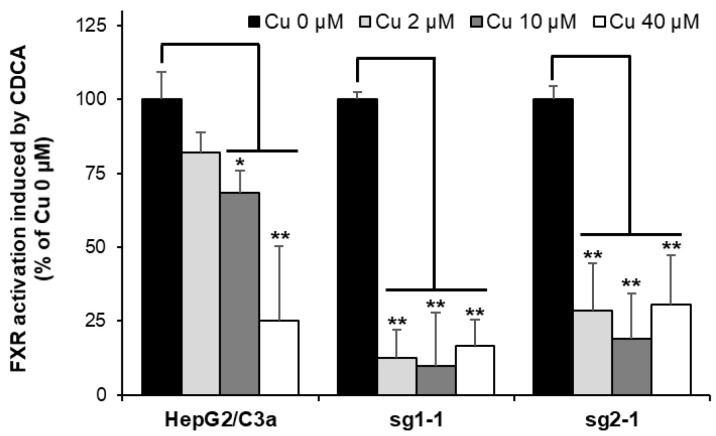
FXR activity impairment in cell lines with or without ATP7B. HepG2/C3a, sg1-1, and sg2-1 cells were exposed to increasing concentrations of Cu and to CDCA. BSEP expression was quantified by qRT-PCR, and the inhibition of FXR was determined by comparing BSEP expression between cells exposed and unexposed to Cu. All the data are expressed as means +/− standard error of the mean of at least three independent experiments. * stands for data statistically different from the control unexposed to Cu with *p* < 0.05, and ** stands for data statistically different from the control unexposed to Cu with *p* < 0.01.

## Data Availability

Data are available upon reasonable request.

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
