# Peer review of "ATP7B-Deficient Hepatocytes Reveal the Importance of Protein Misfolding Induced at Low Copper Concentration"

_cells, 2022, doi:10.3390/cells11213400_

Round 1

Reviewer 1 Report

The manuscript "ATP7B-deficient hepatocytes reveal the importance of protein misfolding induced at a low copper concentration" by Charbonnier et al. represents interesting research regarding the role of Cu in hepatic steatosis, providing a cell line model.

The manuscript is well written. However, the authors should improve the figures' presentation. In particular, the quality of Figure 1A is so low! Besides, I note a sort of confusion in the presentation of figures 1 and 4. In my opinion, they should identify different images better. Regarding other figures, some of them are very crowded (figs 5,6, and 7). The authors should improve their presentation.

The manuscript' English should check by the English mother language reader.

Author Response

The manuscript is well written. However, the authors should improve the figures' presentation. In particular, the quality of Figure 1A is so low! Besides, I note a sort of confusion in the presentation of figures 1 and 4. In my opinion, they should identify different images better. Regarding other figures, some of them are very crowded (figs 5,6, and 7). The authors should improve their presentation.

We thank the reviewer for the positive appreciation of our manuscript. We improved the figure by replacing all figures to higher resolution figures. This should help the reader especially for the crowded figures. Furthermore, for the Figure 1A, we changed the blot to have a higher quality image and also to fulfill reviewer 2 requirement.

Reviewer 2 Report

The present study established the ATP7B-deficient cells as the model of hepatocytes with Wilson disease. ATP7B-deficient cells showed the altered expression of Met, GCLM, HMOX, IL-8 and HSPA6 compared with the original HepG2/C3a. The reviewer considers the present study has the sufficient amount of experiments. However, the reviewer also considers that it is difficult to understand the scientific importance of the present study. The reviewer would like to ask some queries to the authors as described below.

1.    The authors described the fold-change of expression of HMOX, HSPA6 and other factors from line 390 to 442 in the main text. The reviewer can understand the degree of fold-change of those factors is different between ATP7B-deficient cell lines and HepG2/C3a. But the reviewer cannot understand why those data suggests the protein misfolding. In addition, the difference of DNAJ proteins was small compared with other factors. Is there any evidence that the difference of those protein expression means the protein misfolding? The reviewer considers the authors should explain the relation between the expression of those factors and the protein misfolding in the section of discussion with some citations.

2.    The ATP7B-deficient animal model has been already established. In addition, ATP7B-deficient HepG2 has also been established (Pantoom S et al. Cells 10(11), 3118, 2021.). The reviewer would ask to the authors what the novel and important points of the present study are. If the original cell line HepG2/C3a is the point of novelty, the reviewer request to explain the significance of HepG2/C3a as a material of ATP7B-deficient cell and the difference from HepG2.

3.    In line 243-245, the authors described that western-blot analysis confirmed the absence of ATP7B expression in the clones sg1-1, sg2-1 and sg2-2 and the weak expression for sg1-2, while sg1-3 properly expresses ATP7B (Figure 1A). But the reviewer considers there is a weak expression in the lane of sg2-1 according to Figure 1A. The reviewer would request the authors to reconfirm the result of western-blot and to explain why sg2-1, not sg2-2, was selected to use subsequent analyses.

4.    The reviewer considers that the sentences of line 270-272, 331-334, 391-393 need to add some citations.

5.    The reviewer considers that the abstract contains excess description regarding established facts and does not contain the results. The authors should change the style of abstract drastically.

Round 2

Reviewer 2 Report

The reviewer was satisfied with the authors' responses.